# A GABA Receptor Modulator and Semiochemical Compounds Evidenced Using Volatolomics as Candidate Markers of Chronic Exposure to Fipronil in *Apis mellifera*

**DOI:** 10.3390/metabo13020185

**Published:** 2023-01-26

**Authors:** Vincent Fernandes, Kevin Hidalgo, Marie Diogon, Frédéric Mercier, Magaly Angénieux, Jérémy Ratel, Frédéric Delbac, Erwan Engel, Philippe Bouchard

**Affiliations:** 1CNRS, UMR6023, Laboratoire Microorganismes Génome et Environnement, Université Clermont Auvergne, F-63000 Clermont-Ferrand, France; 2INRAE, UR370 Qualité des Produites Animaux, MASS Group, F-63122 Saint-Genès-Champanelle, France

**Keywords:** volatolomics, volatile organic compounds, pesticide, GABA modulator, honeybees

## Abstract

Among the various “omics” approaches that can be used in toxicology, volatolomics is in full development. A volatolomic study was carried out on soil bacteria to validate the proof of concept, and this approach was implemented in a new model organism: the honeybee *Apis mellifera*. Emerging bees raised in the laboratory in pain-type cages were used. Volatolomics analysis was performed on cuticles, fat bodies, and adhering tissues (abdomens without the digestive tract), after 14 and 21 days of chronic exposure to 0.5 and 1 µg/L of fipronil, corresponding to sublethal doses. The VOCs analysis was processed using an HS-SPME/GC-MS method. A total of 281 features were extracted and tentatively identified. No significant effect of fipronil on the volatolome could be observed after 14 days of chronic exposure. Mainly after 21 days of exposure, a volatolome deviation appeared. The study of this deviation highlighted 11 VOCs whose signal abundances evolved during the experiment. Interestingly, the volatolomics approach revealed a VOC (2,6-dimethylcyclohexanol) that could act on GABA receptor activity (the fipronil target) and VOCs associated with semiochemical activities (pheromones, repellent agents, and compounds related to the Nasonov gland) leading to a potential impact on bee behavior.

## 1. Introduction

Toxicology characterizes toxic compounds, studies how they act on organisms, and assesses the associated risks [1]. An organism’s response to a given stress can be understood by studying the biological mechanisms associated with a disturbance [2]. “Omics” approaches represent an alternative to traditional toxicological methods based on a few genes, proteins, or metabolites. Changes in the transcriptome, proteome, or metabolome of an organism may reveal potential markers of exposure to a toxic compound [3,4,5]. When an organism undergoes stress (biotic or abiotic), its metabolism is often modified in order to adapt to disturbances or damages [6,7]. These mechanistic adjustments result in changes in metabolite production, particularly metabolism end products such as Volatile Organic Compounds (VOCs). VOCs are low-molecular-weight compounds and are volatile at room temperature (20 °C) and low pressures (below 10 Pa) [8,9]. The volatolome is defined as the volatile fraction of the metabolome at a given time in a given environment [8]. VOCs can attest to a reorganization of metabolism and thus appear as potential markers of stress exposure [10]. Searching for markers using volatolomics has already been tested in previous work on livestock [10,11,12,13], soil bacteria [7], and gut microorganisms [14,15].

Fipronil (5-amino-1-[2,6-dichloro-4-(trifluoromethyl)-phenyl]-4-trifluotromethyl sulfinyl-lH-pyrazole-3-carbonitrile) is an insecticide of the phenylpyrazole family [16]. It has been introduced in many commercial formulations used in agriculture, veterinary medicine, and as a domestic insecticide. It has a broad toxic spectrum against organisms considered pests such as ants, fleas, ticks, grasshoppers, and locusts. However, in 2004, several products for agricultural use containing fipronil were banned in France (Regent^®^ in particular) owing to the uncertain effects of this substance on humans and the deleterious effects observed in mammals (disturbances of neuronal, hepatic, renal, and thyroid functions) [17,18]. In addition, it has been reported that fipronil has a broad toxic spectrum on vertebrate non-target organisms (such as small mammals, birds, fishes, and amphibians) and on invertebrate ecosystem auxiliaries (such as bees, butterflies, and earthworms) (see [19,20] for a review). Nevertheless, fipronil is still authorized for domestic use and veterinary medicine.

The honeybee, *Apis mellifera*, is a major pollinator insect and participates in the production of more than 30% of the food consumed by humans. However, a decline in its population has been observed, in Europe and North America in particular, over the past three decades [21]. For many years, the honeybee has also been recognized as a sensitive indicator of environmental disturbances. It is an excellent model organism in toxicology due to its role as an ecosystem auxiliary and because there are many routes by which its contamination by pesticides can occur (exposure by contact during spreading, during foraging activities, by contact between bees in the hive, through contaminated food, etc.) [22]. The effects of many toxic compounds including fipronil have been studied in *A. mellifera*. Fipronil is a non-competitive antagonist of γ-aminobutyric acid receptors (also known as GABA-R), and when it binds to these receptors, the flow of chloride ions is blocked in the nerve cells, resulting in hyperexcitability of the nervous system and leading to death [22,23,24]. This property makes fipronil a systemic insecticide highly lethal to bees, with an LD50 of 4.2 ng/bee at 48 h in the case of oral intoxication [19]. Nonetheless, at lower doses, many sub-lethal effects have also been demonstrated at various physiological levels, such as impacts on reproduction [25,26], development [27], foraging activity [28], and the expression of immunity genes [29]. Sub-lethal effects have also been recorded on the intestinal microbiota structure [30] as well as on the establishment of intestinal dysbiosis [31,32,33].

Only a few studies have used volatolomics to assess the impact of a toxic compound on a living organism. It has been assumed that when a bee undergoes pesticide stress, its metabolism is reorganized. This metabolic deviation (focused on VOCs) should allow us to identify potential markers of exposure. This work set out to (i) continue the proof of concept [7], which seeks to promote the volatolomics approach as a relevant tool in the search for xenobiotic exposure markers, (ii) highlight a volatolome deviation following chronic fipronil exposure, and (iii) detect volatile markers that could attest to this exposure. The *A. mellifera* volatolome was analyzed after 14 and 21 days of chronic fipronil exposure in bee abdomens cleared of the digestive tract, and so mainly composed of the cuticle and fat body (i.e., tissues considered to be VOC accumulative compartments), using an HS-SPME/GC-MS approach.

## 2. Materials and Methods

### 2.1. Sampling of Honeybees

The experiment was designed based on previous work demonstrating the impact of an intestinal parasite (*Nosema ceranae*) and fipronil on oxidative balance in bee intestines [33]. Briefly, frames were collected from 3 colonies of the same apiary (UMR 6023, Université Clermont Auvergne, Clermont-Ferrand, France) and placed in the dark at 35 °C and 60% relative humidity. Emergent bees were collected on the frame and randomly distributed in 36 pain-type cages with 50 bees/cage. The bees were fed ad libitum with sucrose syrup complemented with 1% nutritional supplement Provitabee^®^ (Biové Laboratory, Arques, France). To mimic the hive environment, a small piece of PseudoQueen^®^ (Contech Enterprises, Inc., Victoria, BC, Canada) was placed in each cage. The encaged honeybees were then placed in an incubator at 33 °C and 60% relative humidity., A total of 3 experimental conditions, 12 cages/condition, were set up 3 days after emerging: a non-exposed group (control), a 0.5 µg/L fipronil-exposed group (Fip 0.5), and a 1 µg/L fipronil-exposed group (Fip 1). These fipronil concentrations were traditionally used in the laboratory as sublethal doses.

### 2.2. Insecticide Exposure and Artificial Rearing

A stock solution of fipronil (70 g/L) (PESTANAL^®^, Sigma-Aldrich, Darmstadt, Germany) was prepared in dimethyl sulfoxide (DMSO) (Sigma-Aldrich), giving a final concentration of 0.005% DMSO in the feeding syrup. From the beginning of the intoxication with fipronil, the control group was fed with sucrose syrup containing 0.005% DMSO. In the feeding syrups of the Fip 0.5 and Fip 1 conditions, fipronil solution was added to obtain the respective final concentrations of 0.5 µg/L and 1 µg/L.

After 14 days of experimental rearing, bees from 5 cages (for each condition), were sacrificed for the first volatolome analysis. At the end of the experiment, after 21 days of artificial rearing, bees from the 7 remaining cages for each condition were sacrificed for the second volatolome analysis. These 2 sampling points were chosen in accordance with studies previously performed in the laboratory attesting that (i) an increase in mortality appears mainly after 14 days of chronic exposure to fipronil and (ii) bees have a lifespan of about 30 days. The end of the experiment must be set before an excessive increase in natural mortality; 21 days was therefore chosen as a reasonable experimental timeframe.

Mortality was scored daily, and dead bees were removed. Syrup consumption was measured by weighing the feeders containing the nutrient solution every two days.

### 2.3. Volatolomic Analysis

After 14 and 21 days of experimental rearing, the bees were put to sleep using carbon dioxide. For each experimental condition, five bees were randomly sampled from each pain-type cage and dissected to recover their abdomens. Part of the digestive tract was removed (ventricle, ileum, and rectal bulb) by pulling on the sting. The empty abdomens were immersed in liquid nitrogen immediately after sampling and stored at −80 °C. Each sample represented a pool of five abdomens. A total of 4–5 samples per experimental condition (i.e., 6 modalities, 3 exposure concentrations, and 2 exposure durations for a total of 145 bee abdomens) were prepared. The abdomen was chosen for this experiment because it contains most of the insect’s fat body, where many metabolic activities take place, including detoxication activities.

Bee abdomens were ground in liquid nitrogen into a fine homogeneous powder using a stainless steel ball mill (Retsch, Mixer Mill MM200, 2 min at 30 freq/s). About 140 mg of bee abdomen powder was placed in a 10 mL glass vial (Supelco, Sigma-Aldrich) and supplemented with a saturated saline solution (NaCl) at 360 g/L according to the sample weight (V = sample weight/360) to facilitate Volatile Organic Compound trapping. Vials were then sealed under nitrogen flow with magnetic caps with PTFE/silicone septa (Supelco, Sigma-Aldrich) and left for 24 h at 4 °C. VOCs of *A. mellifera* were analyzed using headspace solid-phase microextraction (HS-SPME) coupled to gas chromatography–mass spectrometry (GC-MS) adapted from Bouhlel et al., 2018 [13]. Briefly, the following steps were carried out with an automated sampler (AOC-5000 Shimadzu): (i) preheating of the samples at 40 °C for 10 min in the agitator (500 rpm) and (ii) SPME trapping (with 65 µm of polydimethylsiloxane/divinylbenzene, a 24-gauge needle, and Supelco) of the VOCs for 30 min.

After extraction, thermal desorption was performed at 280 °C for 2 min in the GC inlet. VOC analysis was performed using a GC/MS-full scan (GC2010; QP2010+, Shimadzu, Japan). VOCs were injected in splitless mode into an Rxi^®^-624Sil MS capillary column (60 m × 0.25 mm × 1.4 µm; Restek) with helium as a carrier gas at a flow rate of 1 mL/min. The oven temperature was held at 40 °C for 5 min, ramped up to 230 °C at 3 °C/min, and held at 230 °C for 10 min. The temperature of the transfer line between GC and MS was set at 230 °C. The temperature was set at 180 °C in the MS source and 150 °C in the MS quadrupole. Electron impact energy was set at 70 eV, and data were collected in the range *m*/*z* 33–250 at 10 scans per second.

Peak areas of the VOCs were integrated from the SPME-GC-MS signals using a mass fragment selected for being both specific to the sought-after molecule and free of any co-elution with a home-made automatic algorithm developed by Bouhlel et al., 2018 [13] using Matlab R2017 (The Math Works, Natick, NC, USA). VOCs were tentatively identified from mass spectra via comparison against the NIST 17 mass spectral library (version 1.3, build 4 May 2017) and from retention indices (RI) via comparison with published RI values and those of our internal database. To calculate the experimental RI, a standard alkane solution (Supelco, Sigma-Aldrich) was analyzed with SPME-GC-MS at the end of the analytical campaign.

The fipronil solutions used to prepare the Fip 0.5 and Fip 1 sucrose solutions were also analyzed with the same SPME-GC-MS method to determine whether any of the discriminating VOCs originated from the exposure solution (exogenous compounds).

### 2.4. Data Treatment

#### 2.4.1. Consumption and Survival Monitoring

Survival and consumption data were analyzed with R software (version 2.1.4, http://www.R-project.org, accessed since January 2021). Mortality data were used to build Kaplan –Meier distributions on which a Cox–Mantel test was performed. A Shapiro–Wilk test followed by a Student’s t-test was applied to the consumption data.

#### 2.4.2. Volatolome Analysis

Volatolome data were processed using Statistica (version 12, StatSoft) and R (version 2.1.4) software. A multivariate ANOVA (MANOVA) was performed to observe whether fipronil influenced the bee volatolome. After a Shapiro–Wilk test, discriminant VOCs were identified with a one-way ANOVA. ANOVAs were followed by a Newman–Keuls post hoc procedure among levels of significant factors to detect the origin of the significant differences (a compound was considered discriminant when its signal abundance was significantly different in a certain case/control comparison for a specific exposure duration). A principal component analysis (PCA) was conducted using discriminant VOCs to show the structures of the volatolomic profiles for the six experimental conditions.

## 3. Results

### 3.1. Effect of Fipronil Exposure on Survival and Sucrose Consumption

The survival analysis (Figure 1) showed a 10% decrease in the survival rate for the control condition over the experimental period. For Fip 0.5, a 28% decrease in the survival rate was observed over the 21 days, while a 68% decrease in the survival rate was found for the Fip 1 group. For this condition, a maximum decrease of 14% was observed between day 13 and day 14. It was in this period that we observed a marked disparity between the Fip 1 distribution and the Fip 0.5 and control distributions. Indeed, for the Fip 1 condition, there was a strong decrease in the survival rate after the first 14 days of treatment. What was observed on the survival curves was confirmed with a Cox–Mantel test (*p*-value < 0.05) and could be determined between each survival distribution. Fipronil, at the doses tested with chronic exposure in the experiment, therefore had a significant impact on the survival of *A. mellifera*.

An average daily intake of 35 mg of sucrose/bee/day was observed for the control group, 32 mg of sucrose/bee/day for the Fip 0.5 group, and 31 mg of sucrose/bee/day for the Fip 1 group (Figure 2). Bees consumed 0.013 ng of fipronil/bee/day (LD50/323) and 0.026 ng of fipronil/bee/day (LD50/161) for the Fip 0.5 and Fip 1 groups, respectively (the data are not shown). At the end of the experiment, the fipronil consumption was still below the LD50, with a mean cumulated consumption of 0.273 ng/bee (LD50/15.4) for the Fip 0.5 group and 0.546 ng/bee (LD50/7.7) for the Fip 1 group. No significant effect of the treatments on consumption was observed based on the Student’s *t*-test (*p*-value > 0.05). Chronic fipronil exposure did not, therefore, affect the daily food intake of *A. mellifera*.

### 3.2. Volatolome Analysis of A. mellifera Exposed to Fipronil

In all, 281 features were extracted in this study and treated with a MANOVA. It showed a significant effect of fipronil concentration, the duration of treatment, and the interaction between these two experimental factors on the A. mellifera volatolome (*p*-values < 0.05, Table 1). Among the features, fourteen were tentatively identified because they were considered discriminant according to a one-way ANOVA and Newman–Keuls post hoc procedure (their signal abundance was modulated in at least one case/control comparison for a specific exposure duration, as shown in Table 2). Three unidentified compounds (among the fourteen mentioned above) showed a significantly modulated signal abundance.

Based on discriminant VOCs, a PCA was built (Figure 3) to visualize the structuring of the dataset. The first PCA axis (PC1) accounted for 59.5% of the dataset inertia and showed a separation between the Fip 1 group at 21 days and the other experimental conditions. The second PCA axis (PC2) explained 15.6% of the dataset inertia and showed a separation between the control and Fip 0.5 groups at 21 days and the group containing all conditions after 14 days of exposure. No separation was observed between the 3 experimental conditions at 14 days, so there was little or no difference in the volatolomic profiles after 14 days of treatment. There was little difference between the volatolomic profiles of the control and Fip 0.5 groups at 21 days, but a trend was discernable because Fip 0.5 was slightly out of step with the control group. On the other hand, the Fip 1 group at 21 days stood out clearly from these other 2 conditions. This suggests that there is an exposure threshold below which the overall volatolomic response is not yet significant.

The correlation circle (Figure 4) based on the fourteen discriminant VOCs and boxplots (Figure 5 and Figure 6) representing the evolution of signal abundance separate the compounds into two clusters: (i) those whose signal abundance evolves with fipronil concentrations (in particular with the Fip 1 condition at twenty-one days) and (ii) those whose signal abundance may evolve with the exposure duration. The eleven discriminant VOCs identified (excluding the three unknown compounds) in bees exposed to fipronil could also be divided into four different groups according to their putative functions (Table 2).

## 4. Discussion

Fipronil is a systemic insecticide acting on the gamma-aminobutyric acid receptor (GABA-R) leading to the closure of chloride channels and a persistent nerve signal. Among the identified VOCs, 2,6-dimethylcyclohexanol is described in the literature as a GABA-R modulator [34]. This compound binds to GABA-R on a different binding site than GABA and via an allosteric effect causes a prolonged opening of chloride channels. We can assume that this VOC acts as a modulator produced by the organism to offset or to counteract the fipronil effects. The signal abundance of this compound evolves in the presence of fipronil after 21 days of exposure. These results suggest that there is a potential modulation of GABA-R. This detected VOC could thus be considered a potential biomarker of neural disturbance resulting from exposure to fipronil. In this study, we worked on VOCs arising from the abdominal cuticle and fat body. However, 2,6-dimethylcyclhexanol is a neurotropic compound. In each sample preparation, the intestinal tract was removed to keep the fat body and abdominal cuticle, but the abdominal neural ganglia chain (attached to the inner side of the cuticle) was preserved. This VOC was probably not extracted from the cuticle or fat body but was produced and/or stored in the abdominal nerve ganglia.

The presence of xenobiotics in the close environment of an organism causes stress. In social organisms, it results in metabolism modulations and/or the production and dissemination of intra- or interspecific communication molecules that cause a change in the behavior of neighboring organisms. These kinds of molecules are called semiochemical compounds. In this work, 10 discriminant compounds could be related to pheromones, alarm pheromones, repellent agents (allomones), or molecules related to the Nasonov gland (this is located under the abdominal cuticle in the dorsal position and plays a role in orientation and swarming). 1-nonen-3-ol and 9-decenol are described in the literature [35] as repellent agents. In this study, their signal abundance evolved according to the fipronil exposure at 1 µg/L at 21 days. Three VOCs (2-decen-1-ol, 2-heptanone, and 3-nonanone) are described as alarm pheromones or pheromones in some Hymenoptera and beetles [36,37,38,39,40,41,42], whose signal abundance changed under a Fip 1 condition at 21 days. Five VOCs (farnesol, geranyl acetone, nerolic acid, geranic acid, and 6-methyl-5-hepten-2-one) are described as molecules related to the Nasonov gland in A. mellifera (Figure 7) [43,44,45] or citral degradation [46]. In this study, their signal abundance changed with fipronil exposure at 21 days. It is interesting that an almost complete Nasonov metabolic pathway scheme could be reconstructed from volatolome data. It highlights molecules with semiochemical activities (nerolic acid, geranic acid, and farnesol) and some synthesis or degradation intermediates (geranyl acetone and 6-methyl-5-hepten-2-one). These compounds may then be placed in a metabolic context indicating that fipronil exposure is associated with the role of the Nasonov gland (i.e., behavioral fitness).

This prospective analysis of the role and origin of VOCs shows that chronic fipronil poisoning alters the exchange of information between organisms. It is therefore necessary to link these results to behavioral and social observations in bees. Further experiments on colonies exposed or not to fipronil could study general bee behavior (aggressiveness, orientation, flight time, and foraging activity) and define the link between the VOCs identified here and bee behavior. As in the case of the neurotropic VOC, not all these semiochemical compounds seem to come from the abdominal cuticle or fat body; some come from various subcuticular glands (the Nasonov gland in particular) that could not be removed during sample preparation.

The great majority of VOCs highlighted in this study have semiochemical activities. We can consider that VOCs accumulated in bee abdomens are not the consequence of metabolism reorganization in response to a xenobiotic, but rather an accumulation of communication signals consecutive to the perturbation caused by fipronil. In this study, VOCs were recorded over a long experimental period (14 and 21 days). To record a volatolome deviation consecutive to a metabolism modulation, it may be necessary to perform the same analyses over short time periods, 24 or 48 h, for example. The choice of organism tissue is also of great importance: extracting VOCs from the head would have been a better way to link fipronil and counteracting VOCs.

Nonetheless, the characterization of VOCs recovered from a small number of bees offers an obvious way to monitor xenobiotic exposure in beehives regarding neuroreceptor-associated molecules and pheromones. Clearly, we are unable to select the relevant biomarkers; more data are needed in both the laboratory and the open field. Our findings contribute to the proof of concept of volatolomics and help to understand the metabolic changes associated with pesticide poisoning in honeybees.

## Figures and Tables

**Figure 1 metabolites-13-00185-f001:**
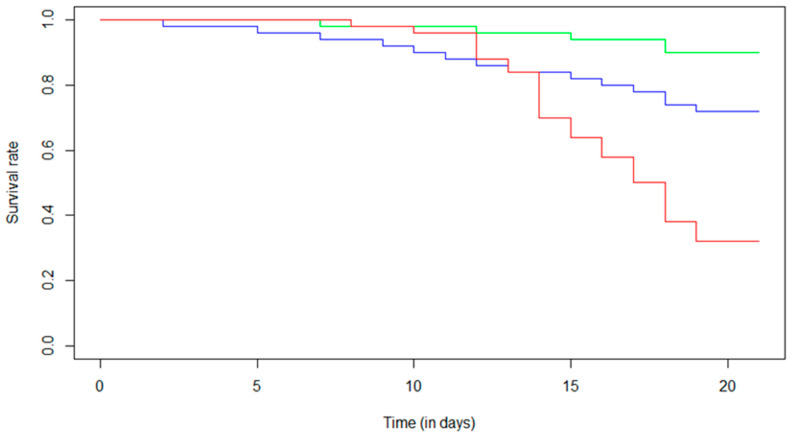
Effect of chronic exposure at different fipronil concentrations on the survival of A. mellifera. The Kaplan–Meier distribution shows the survival proportions of bees (i) not exposed to fipronil (control, green line), (ii) exposed to 0.5 µg/L of fipronil (Fip 0.5, blue line), and (iii) exposed to 1 µg/L of fipronil (Fip 1, red line). The Cox–Mantel test shows a significant decrease in survival of bees exposed to both fipronil concentrations compared with control group (*p*-value = 0.020 for Fip 0.5 and *p*-value ≤ 0.0001 for Fip 1). In addition, a significant difference was observed between the two exposed groups (*p*-value = 0.0003).

**Figure 2 metabolites-13-00185-f002:**
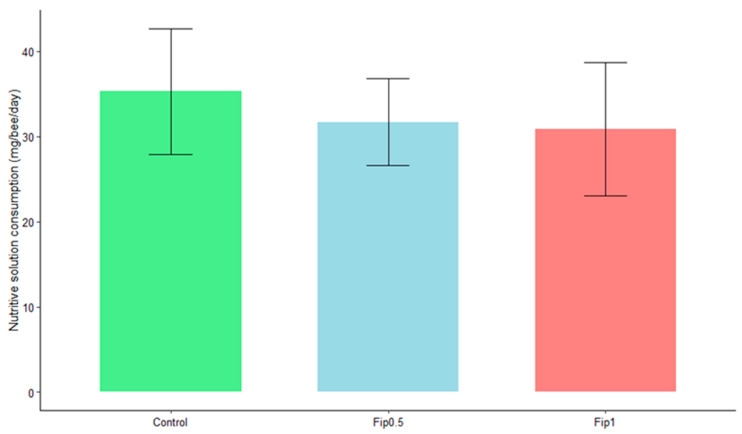
Average daily consumption by bees of a 50% sucrose solution under different conditions. Statistical analysis with Student’s t-test did not show a significant effect of fipronil on daily sugar consumption between the different conditions (*p*-value ˃ 0.05).

**Figure 3 metabolites-13-00185-f003:**
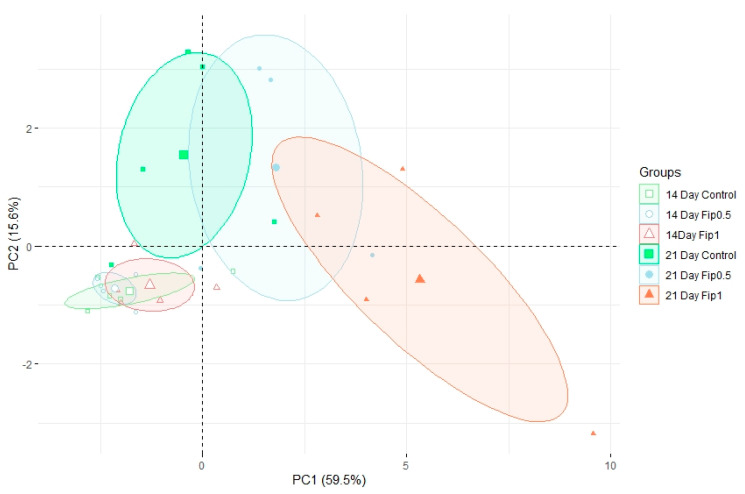
Normed PCA showing the volatolomic profiles in A. mellifera exposed or not to different fipronil concentrations. For each of the 6 conditions, 4−5 pools of 5 abdomens were analyzed. For each group, the large icon (specific to each group) represents the barycenter of the group. The ellipses are 95% confidence ellipses, which means that in 95% of cases, a point belonging to a group will be found inside the ellipse of that group.

**Figure 4 metabolites-13-00185-f004:**
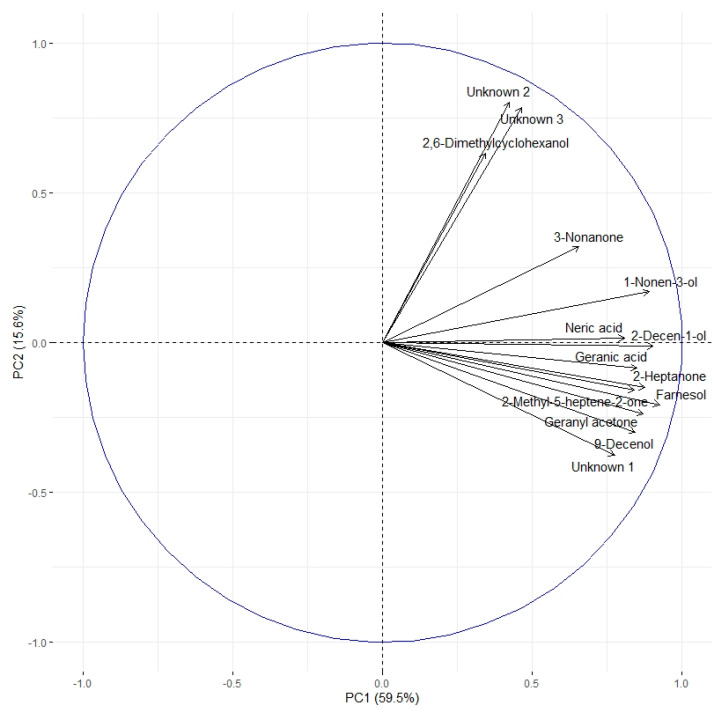
Correlation circle based on the 14 discriminating VOCs. Two clusters can be identified. The first contains 11 of the 14 discriminating VOCs. The second one contains three compounds (2,6-dimethylcyclohexanol and unknown compounds 2 and 3). Considering the evolution of the signal abundance, the first cluster can be related to the fipronil exposure and the second cluster to the exposure duration.

**Figure 5 metabolites-13-00185-f005:**
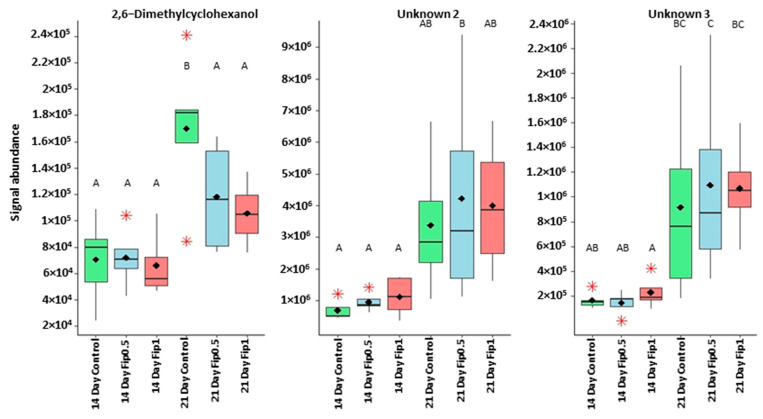
Boxplots representing discriminant compounds whose signal abundance increases with treatment duration irrespective of fipronil concentration. Black diamonds represent the average signal abundance of each group. The black horizontal crossbars represent the medians. Red asterisks represent outlier values. Differences in signal abundance are represented by letters. When 2 conditions are characterized by the same letter, there is no significant difference between them (*p*-value > 0.05, according to a Newman–Keuls test).

**Figure 6 metabolites-13-00185-f006:**
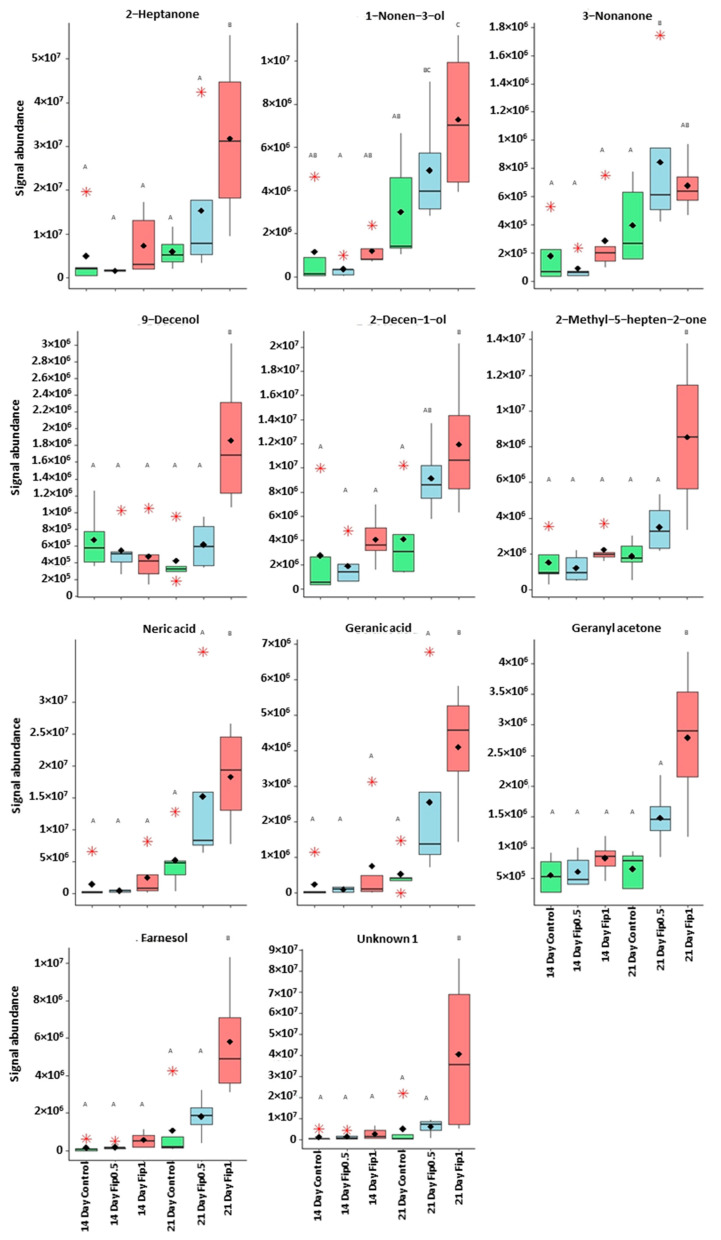
Boxplots representing discriminant compounds whose signal abundance increased with fipronil concentration. Black diamonds represent the average signal abundance of each group. The black horizontal crossbars represent the medians. Red asterisks represent outlier values. Differences in signal abundance are represented by letters. When 2 conditions are characterized by the same letter, there is no significant difference between them (*p*-value > 0.05, according to a Newman–Keuls test).

**Figure 7 metabolites-13-00185-f007:**
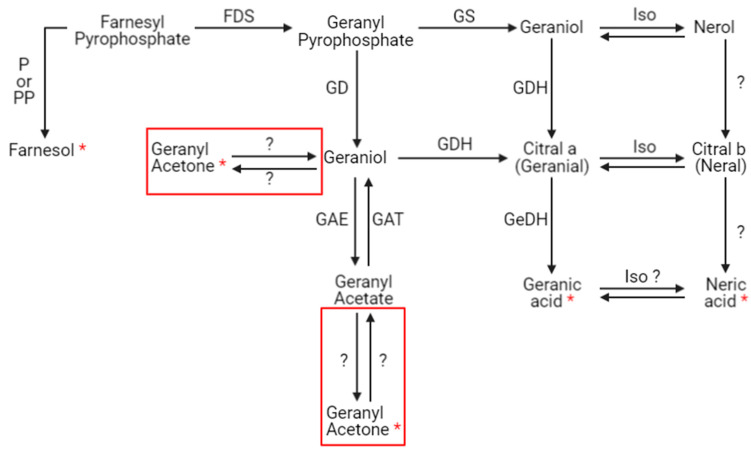
Putative Nasonov gland metabolic pathway. Seven compounds produced in the Nasonov gland are known to have semiochemical activities: farnesol, geraniol, nerol, geranial, neral, geranic acid, and neric acid. Question marks represent hypothetical transformations. Red frames containing geranyl acetone represent these two hypothetical synthesis pathways. P = phosphatase, PP = pyrophosphatase, FDS = farnesyl diphosphate synthase (or farnesyl pyrophosphate synthase), GD = geranyl diphosphatase (or geranyl pyrophosphatase), GAE = geranyl acetate esterase, GAT = geranyl acetyl transferase, GS = geraniol synthase, GDH = geraniol dehydrogenase, Iso = isomerase, and GeDH = geranial dehydrogenase (according to [43,44,45,47,48,49]). Red stars represent compounds related to the Nasonov gland pathway identified in the volatolome analysis.

**Table 1 metabolites-13-00185-t001:** Results of the MANOVA.

Factor	Df	F-Value	*p*-Value
Concentration	2	6.24	0.005
Treatment duration	1	25.89	0.01
Concentration × Treatment duration	2	8.16	0.002

**Table 2 metabolites-13-00185-t002:** VOCs whose signal abundance was significantly modulated in A. mellifera after 14 or 21 days of chronic exposure to fipronil. The putative identity is based on the comparison of the results obtained with the GC-MS Solution software (version 4.11 SU2, Shimadzu) and the AMDIS (Automated Mass-Spectral Deconvolution and Identification System) software (version 2.72) of the National Institute of Science and Technology (NIST). The retention times (RT) indicated correspond to those obtained with the GC column Rxi^®^—624Sil MS fused silica column (60 m × 0.25 mm × 1.4 µm, Restek). The mass/charge ratios (*m*/*z*) correspond to the mass fragments used for area determination. The RI NIST values correspond to the retention indices reported with a D85 capillary column in the NIST database. The function proposed for each molecule is based on searches in bibliographic databases and in specialized databases such as The Pherobase, which is specialized in compounds with semiochemical activities (https://www.pherobase.com/, accessed since January 2021).

Putative Identity	*m*/*z*	RT	RI NIST	RI Calculated	Proposed Function
2,6dimethylcyclohexanol	71	40.8	1112	1133	Positive modulator of GABA receptors
1-nonen-3-ol 9-decenol	55	38.5	1078	1079	Repellent agent
	55	44.5	1256	1261	
2-decen-1-ol 2-heptanone 3-nonanone	57	45.0	1285	1285	Pheromones and alarm pheromones
	43	27.0	891	898	
	43	38.9	1087	1087	
Farnesol	69	54.1	1713	NA	Compounds related to the Nasonov gland
Geranyl acetone	69	48.0	1460	1455	
Neric acid	69	45.5	1340	1312	
Geranic acid	69	46.1	1345	1345	
6-methyl-5-hepten-2-one	43	32.0	986	985	

## Data Availability

Data is not publicly available due to privacy or ethical restrictions. If you want more information’s or have an access to the data used in this paper, please contact the corresponding authors using their email address: erwan.engel@inrae.fr and philippe.bouchard@uca.fr.

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
