# Peer review of "A GABA Receptor Modulator and Semiochemical Compounds Evidenced Using Volatolomics as Candidate Markers of Chronic Exposure to Fipronil in Apis mellifera"

_metabolites, 2023, doi:10.3390/metabo13020185_

Round 1

Reviewer 1 Report

The authors want to identify semiochemical compounds evidenced by volatolomics as potential markers of Apis mellifera chronic exposure to xenobiotic stress. This hypothesis is interesting and very important to know the metabolism modulations and/or production and dissemination of intra- or interspecific communication molecules that will lead to a change in be havior of social insects. The authors presented molecules characterization represent an obvious way to monitor xenobiotic exposition in beehives regarding neuroreceptors associated molecules and pheromones,even they are unable at this point of the experiment to select the relevant biomarker. But the great relevance towards the volatolomic proof of concept of the metabolic changes associated with pesticides poisoning in honeybees will be clearly clarified in furfure. Depend on the importance of this manuscript, it is suitable for publication, also the experiments were well taken and data was carefully analyzed.  

Author Response

Vincent FERNANDES

Université Clermont Auvergne

UMR CNRS Laboratoire Microorganismes Génome et Environnement

Campus Universitaire des Cézeaux, 1 Impasse Amélie Murat, 63170 Aubière, France

Reviewer #1

Metabolites

Jan, 13, 2023

Dear Reviewer #1,

            We would like to thank you for providing your comments on our manuscript.

            As we did not have any specific request from you, we take the liberty of using this “point-by-point responses” to forward you the “authors general comments” that have been communicated to the other reviewers.

Authors general comments: We would like to inform you that, following a request of the reviewer #3, the first version of the manuscript has been proofread by a specialized translator (made by Richard RYAN, scientific and technic translator), many modifications have been made on the linguistic and stylistic aspects, these changes are not presented here but are accessible using the “Track Changes” function of the Microsoft Word attached document.

In addition, we would like to point out here that an error was present in the materials and methods section (2.4. Data treatment; 2.4.2. Volatolome analysis) of the first version of the manuscript. It was mentioned that two-way ANOVAs were performed, whereas one-way ANOVAs were carried out. We have renewed our statistical analysis in their entirety to ensure the quality of the candidate markers. As a result, we have choosing to exclude two candidate markers (2,3-butanediol and citral) that did not fully meet the set criteria. This has led to the modifications of the figures 3, 4, 6, and 7, and the table 1. This material and method section has been also rewritten and necessary modifications were brought to the results and discussion sections.

            We have revised our manuscript according to the majority of the reviewers’ comments. The changes can be highlight in the Microsoft Word attached manuscript using the “Track Changes” function.

            Thank you again for the time you took to study our manuscript. We look forward to hearing you.

Best regards,

Vincent FERNANDES.

Reviewer 2 Report

General comments:

The article “GABA receptor modulator and semiochemical compounds evidenced by volatolomics as candidate markers of Apis mellifera chronic exposure to fipronil” has evidenced the possibly act on the GABA receptors activity and VOCs that are associated to semiochemical activities leading to a potential impact on bee behavior. I have a few suggestions to improve it.

Introduction

First paragraph is too long.

I would recommend stating the hypothesis in the introduction.

Material and Methods: It seems that the analyzes have been conducted appropriately; however, it is necessary to show that the assumptions of ANOVA and MANOVA have been attempted.

Results:

In my opinion the results have been well described.

Discussion:

The first paragraph of the discussion could be transferred to introduction. It is not a discussion.

The paragraph between the lines 311-346 is too long.

Author Response

Vincent FERNANDES

Université Clermont Auvergne

UMR CNRS Laboratoire Microorganismes Génome et Environnement

Campus Universitaire des Cézeaux, 1 Impasse Amélie Murat, 63170 Aubière, France

Reviewer #2

Metabolites

Jan, 13, 2023

Dear Reviewer #2,

            We would like to thank you for providing your comments on our manuscript.

            To begin this “point-by-point responses” we allow ourselves to forward you the “authors general comments” that have been communicated to the other reviewers.

Authors general comments: We would like to inform you that, following a request of the reviewer #3, the first version of the manuscript has been proofread by a specialized translator (made by Richard RYAN, scientific and technic translator), many modifications have been made on the linguistic and stylistic aspects, these changes are not presented here but are accessible using the “Track Changes” function of the Microsoft Word attached document.

In addition, we would like to point out here that an error was present in the materials and methods section (2.4. Data treatment; 2.4.2. Volatolome analysis) of the first version of the manuscript. It was mentioned that two-way ANOVAs were performed, whereas one-way ANOVAs were carried out. We have renewed our statistical analysis in their entirety to ensure the quality of the candidate markers. As a result, we have choosing to exclude two candidate markers (2,3-butanediol and citral) that did not fully meet the set criteria. This has led to the modifications of the figures 3, 4, 6, and 7, and the table 1. This material and method section has been also rewritten and necessary modifications were brought to the results and discussion sections.

            We have revised our manuscript according to the majority of the reviewers’ comments. The changes can be highlight in the Microsoft Word attached manuscript using the “Track Changes” function. In this point-by-point responses, our responses are given in italics below the reviewer comments, the text from the first version written in red and the text corresponding to this revised version written in blue.

1-Introduction:

1a-First paragraph is too long.

Authors: We agree and have revised this paragraph (lines 28 to 49) by the following modifications:

                - The lines 28 to 31 were rewritten: Toxicology is the study of the toxic effect of chemicals on organisms. Usually, this field is based on the characterization of a toxic compound, its mode of action on organisms and the assessment of the associated risk [1]. Determining the biological mechanisms associated with a disturbance is the key to study organism’s response to a given stress [2] was transformed to “Toxicology characterizes toxic compounds, studies how they act on organisms, and assesses the associated risks [1]. An organism’s response to a given stress can be understood by studying the biological mechanisms associated with a disturbance [2]”.

                -The lines 35 to 38 were deleted (“Thus, it opens a large field for new markers that may be monitored using conventional experiments. The objective is not anymore to characterize exposure by quantifying pesticides residues (or their degradation products) but to highlight makers testifying to this exposure on organisms belonging a trophic chain.”).

                -The suggestions made by the translator during the proofreading also made it possible to lighten this first paragraph.

1b-I would recommend stating the hypothesis in the introduction.

Authors: We agree with the reviewer and have modified the last paragraph of the introduction (lines 82 to 88): “Only few studies have used volatolomics to assess the impact of a toxic compound on a living organism. The objectives of this work was determined to (i) highlight a volatolome deviation following fipronil chronic exposure and to (ii) detect volatile markers that can be testify to this exposure. Apis mellifera volatolome was analyzed after 14 and 21 days of fipronil chronical exposure from bees’ abdomen cleared of the digestive tract, thus mainly composed in cuticle and fat body (i.e. considered as VOCs accumulative compartments), using an HS-SPME/GC-MS approach” was transformed to “Only few studies have used volatolomics to assess the impact of a toxic compound on a living organism. It has been assumed that when the bee undergoes pesticide stress, its metabolism will be reorganized. This metabolic deviation (focused on VOCs) should al-low identify potential markers of exposure. This work set out to (i) continue the proof of concept [7] which seeks to promote the volatolomics approach as a relevant tool in the search for xenobiotic exposure markers, (ii) highlight a volatolome deviation following chronic fipronil exposure and, (iii) detect volatile markers that could attest to this exposure. The Apis mellifera volatolome was analyzed after 14 and 21 days of chronic fipronil expo-sure from bee abdomens cleared of the digestive tract, and so mainly composed of cuticle and fat body (i.e., considered as VOCs accumulative compartments), using an HS-SPME/GC-MS approach”.

2-Materials and Methods: It seems that the analyzes have been conducted appropriately; however, it is necessary to show that the assumptions of ANOVA and MANOVA have been attempted.

Authors: Thank you for this relevant remark. Indeed, the hypotheses of normality and homoscedasticity were made before the statistical analysis. Due to the structure of our dataset, we followed the prescribed assumptions: normality is mandatory but it is not advisable to evaluate the homoscedasticity. We therefore carried out a Shapiro-Wilk test before performing the analyses. In this revised version of the manuscript we have added a mention of this test in the corresponding materials and methods sections (lines 174-175): “After a Shapiro-Wilk test, discriminant VOCs were identified by one-way ANOVA”.

3-Results: In my opinion the results have been well described.

Authors: Thank you for this comment.

4-Discussion:

4a- The first paragraph of the discussion could be transferred to introduction. It is not a discussion.

Authors: We agree with the reviewer and have deleted the first paragraph of the discussion (lines 281 to 286). The first part of this paragraph was added to the last paragraph of the introduction (see comment 1b) and the second part was introduced on the section Materials and Methods subsection 2.3. Volatolomic analysis (line 147 and following): “The abdomen was chosen for this experiment because it contains most of the insect’s fat body, where many metabolic activities take place, including detoxication activities”.

4b-The paragraph between the lines 311-346 is too long.

Authors: Thank you for this remark. We have revised this paragraph according to the following items:

                -The initial paragraph was split in two distinct parts: lines 311 to 338 and lines 338 to 346.

                -The lines 322 to 329 were deleted (“Recently, Saccà et al., 2021 [43] described that 2-heptanone is produced by bacteria which are associated with bees and beehives. This specific VOC could also act has antiparasitic molecule against Varroa destructor. It is interesting to mind that VOCs observed from pesticide exposure may be produced by the bee’s microbiota from surface body or intestine. During VOCs recovery step from insect, the whole abdominal intestine is removed in or-der to avoid contamination with bacterial VOCs. It is not known if bacterial VOCs could be stored in insect fat body”).

                - The suggestions made by the translator during the proofreading also made it possible to lighten this paragraph.

            Thank you again for the time you took to study our manuscript. We look forward to hearing you.

Best regards,

Vincent FERNANDES.

Reviewer 3 Report

The authors present work to verify that the volatolome can be used to understand semiochemical changes and possible signaling in social insects. The use of an insecticide as a trigger for the stress response and measuring the insects VOCs is an interesting approach. Overall, the information presented is interesting but the English editing is distracting from the overall message of the paper. 

Specific comments:

Lines 36-38: Phrasing is odd. Consider revising. I'm not sure I understand the meaning of the sentence.

Introduction: There are a considerable amount of instances where words need to be reorganized in sentences to conform with English grammar. I think all of the information is present for the background of the paper, but the English is distracting.

Figure 1. A legend on the graph with the corresponding colors of lines would be helfpul.

Figure 4. Which cluster goes to what abundance.

Author Response

Vincent FERNANDES

Université Clermont Auvergne

UMR CNRS Laboratoire Microorganismes Génome et Environnement

Campus Universitaire des Cézeaux, 1 Impasse Amélie Murat, 63170 Aubière, France

Reviewer #3

Metabolites

Jan, 13, 2023

Dear Reviewer #3,

            We would like to thank you for providing your comments on our manuscript.

            To begin this “point-by-point responses” we allow ourselves to forward you the “authors general comments” that have been communicated to the other reviewers.

Authors general comments: We would like to inform you that, following a request of the reviewer #3, the first version of the manuscript has been proofread by a specialized translator (made by Richard RYAN, scientific and technic translator), many modifications have been made on the linguistic and stylistic aspects, these changes are not presented here but are accessible using the “Track Changes” function of the Microsoft Word attached document.

In addition, we would like to point out here that an error was present in the materials and methods section (2.4. Data treatment; 2.4.2. Volatolome analysis) of the first version of the manuscript. It was mentioned that two-way ANOVAs were performed, whereas one-way ANOVAs were carried out. We have renewed our statistical analysis in their entirety to ensure the quality of the candidate markers. As a result, we have choosing to exclude two candidate markers (2,3-butanediol and citral) that did not fully meet the set criteria. This has led to the modifications of the figures 3, 4, 6, and 7, and the table 1. This material and method section has been also rewritten and necessary modifications were brought to the results and discussion sections.

            We have revised our manuscript according to the majority of the reviewers’ comments. The changes can be highlight in the Microsoft Word attached manuscript using the “Track Changes” function. In this point-by-point responses, our responses are given in italics below the reviewer comments, the text from the first version written in red and the text corresponding to this revised version written in blue.

1-Introduction:

1a- There is a considerable amount of instances where words need to be reorganized in sentences to conform with English grammar. I think all of the information is present for the background of the paper, but the English is distracting.

Authors: Thank you for this remark, we agree with the it. As a result, the first version of the manuscript has been proofread by a specialized translator (Richard RYAN, scientific and technic translator), many modifications have been made on the linguistic and stylistic aspects, these changes are not presented here but are accessible using the “Track Changes” function of the Microsoft Word attached document.  

1b- Lines 36-38: Phrasing is odd. Consider revising. I'm not sure I understand the meaning of the sentence.

Authors: We thank you for pointing out these lines. Following the request of the reviewer #2, the first paragraph of the introduction has been reduced, we have chosen to remove the lines 35 to 38 (“Thus, it opens a large field for new markers that may be monitored using conventional experiments. The objective is not anymore to characterize exposure by quantifying pesticides residues (or their degradation products) but to highlight makers testifying to this exposure on organisms belonging a trophic chain.”) to facilitate the reading.

2- Figure 1: A legend on the graph with the corresponding colors of lines would be helpful.

Authors: We thank you for this proposition. For this first figure we have chosen to put all essential information in the legend. Indeed, the legend introduces the color code that will be followed throughout this paper. As a result, we have chosen to not add the names of the different Kaplan-Meier distributions directly on the figure, the objective being that the reader focuses first on the legend of this figure.

3- Figure 4: Which cluster goes to what abundance.

Authors: Thank you for this relevant remark. We have added these lines “Two clusters can be identified. The first contains 11 of the 14 discriminating VOCs. The second one contains three compounds (2,6-dimethylcyclohexanol and unknown compounds 2 and 3). Considering the evolution of the signal abundance, the first cluster can be related to the fipronil exposure, the second cluster to the exposure duration” to the legend of the figure 4 (line 263 and following).

            Thank you again for the time you took to study our manuscript. We look forward to hearing you.

Best regards,

Vincent FERNANDES.

Round 2

Reviewer 3 Report

Much improved. Thank you to the authors for taking the time to improve this manuscript as it has valuable information within.